# Assessing Ebola virus circulation in the Tshuapa province (Democratic Republic of the Congo): A One Health investigation of wildlife and human interactions

Rianne van Vredendaal[1]ʘ*, Léa Joffrin[1]ʘ, Antea Paviotti[2,3], Claude Mande[4,5], Solange Milolo[6], Nicolas Laurent[7], Léa Fourchault[1,7], Douglas Adroabadrio[4], Pascal Baelo[4], Steve Ngoy[4], Papy Ansobi[8], Casimir Nebesse[4,5], Martine Peeters[9], Ahidjo Ayouba[9], Maeliss Champagne[9], Julie Bouillin[9], Jana Těšíková[1], Natalie Van Houtte[1], Sophie Gryseels[1,7], Maha Salloum[2,3], Freddy Bikioli[6], Séverine Thys[10], Jimmy Mpato[11], Ruben Ilonga[12], Henri Kimina[13], Ynke Larivière[2,3], Gwen Lemey[2,3], Pierre Van Damme[2], Jean-Pierre Van Geertruyden[3], Hypolite Muhindo-Mavoko[6], Patrick Mitashi[6], Herwig Leirs[1], Erik Verheyen[1,7], Guy-Crispin Gembu[4,5], Joachim Mariën[1,14]

1 Evolutionary Ecology Group, Department of Biology, University of Antwerp, Wilrijk, Belgium, 2 Centre for the Evaluation of Vaccination, Vaccine and Infectious Disease Institute, University of Antwerp, Wilrijk, Belgium, 3 Global Health Institute, Department of Family Medicine and Population Health, University of Antwerp, Wilrijk Belgium, 4 Centre for Biodiversity Monitoring, University of Kisangani, Kisangani, Democratic Republic of Congo, 5 Department of Ecology and Wildlife Management, University of Kisangani, Kisangani, Democratic Republic of Congo, 6 Tropical Medicine Department, University of Kinshasa, Kinshasa, Democratic Republic of Congo, 7 Vertebrate Group, Directorate Taxonomy and Phylogeny, Royal Belgian Institute of Natural Sciences, Brussels, Belgium, 8 Ecology and Control of Infectious Diseases Unit, Basic Sciences Department, University of Kinshasa, Kinshasa, Democratic Republic of Congo, 9 TransVIHMI, University of Montpellier, Institute for Research and Sustainable Development (IRD), INSERM, Montpellier, France, 10 ASTRE, French Agricultural Research Centre for International Development (CIRAD), Montpellier, France, 11 Boende General Hospital of Reference, Boende, Democratic Republic of Congo, 12 Central Office of Boende Health District, Boende, Democratic Republic of Congo, 13 Anthropology Department, University of Kinshasa, Kinshasa, Democratic Republic of Congo, 14 Virus Ecology Group, Department of Biomedical Sciences, Institute of Tropical Medicine, Antwerp, Belgium

ʘ These authors contributed equally to this work.
* rianne.vv@outlook.com

## Abstract

The wildlife reservoir and spillover mechanisms of Ebola virus remain elusive despite extensive research efforts in endemic areas. This study employed a One Health approach to examine the virus' circulation in wildlife and the associated human exposure risks in the Tshuapa province of the Democratic Republic of the Congo. We screened 1049 samples from 919 animals, predominantly small mammals, collected in 2021, and 380 samples from inhabitants of Inkanamongo village, the site of an Ebola virus disease outbreak in 2014. These samples were screened for evidence of current (RNA) or past (IgG antibodies) Ebola virus infections. We also conducted interviews with 167 individuals in the surrounding districts to assess their interactions with wildlife. While no Ebola virus RNA was detected in the wildlife

reproduction in any medium, provided the original author and source are credited.

**Data availability statement:** The cytochrome b sequences generated for host identification in this study have been deposited in the European Nucleotide Archive (ENA) at EMBL-EBI under accession number PRJEB101710 (https://www.ebi.ac.uk/ena/browser/view/PRJEB101710). The study also included a subset of cytochrome b sequences from ENA project PRJEB100689, corresponding to accessions OZ348072-OZ348122 (https://www.ebi.ac.uk/ena/browser/view/PRJEB100689). All other relevant data are within the manuscript and its Supporting information files.

**Funding:** This research was funded through the BIODIV-AFREID project (https://www.biodiversa.eu/2022/10/31/biodiv-afreid/) in the 2018-2019 BiodivERsA joint call for research proposals under the European Union's Horizon 2020 BiodivERsA3 ERA-Net COFUND program (grant number ANR-19-EBI3-0004 to HL [https://cordis.europa.eu/project/id/642420]), the EBO-SURSY project (https://ebo-sursy.woah.org/) and the EBOVAC3 project (https://www.ebovac.org/ebovac-3/). The EBO-SURSY project was funded by the European Union (FOOD/2016/379-660 to MP [https://research-and-innovation.ec.europa.eu/index_en]). The EBOVAC3 project has received funding from the Innovative Medicines Initiative 2 (IMI2) Joint Undertaking under grant agreement 800176 (to PVD [www.imi.europa.eu]). This joint undertaking receives support from the European Union's Horizon 2020 research and innovation program (IMI2 grant number 800176 to PVD [https://research-and-innovation.ec.europa.eu/index_en]), the European Federation of Pharmaceutical Industries and Associations (IMI2 grant number 800176 to PVD [https://www.efpia.eu/]), and the Coalition for Epidemic Preparedness Innovations (INAN1901 to PVD [https://cepi.net/]). Additional funding for DNA barcoding was provided by the Research Foundation – Flanders (FWO) projects (G051322N to SG and G054820N to EV [https://www.fwo.be/en/]). PVD and HL are PI in the UAntwerp Center of Excellence VAX-IDEA. LJ was a postdoctoral fellow of the Research Foundation–Flanders (FWO) (grant number 1271922N [https://www.fwo.be/en/]). The funders had no role in study design, data collection and analysis, decision to publish, or preparation of the manuscript.

**Competing interests:** The authors have declared that no competing interests exist.

samples, anti-orthoebolavirus IgG antibodies were found in 13 bats and 38 rodents. Among the human participants, 120 individuals had IgG antibodies against at least 1 orthoebolavirus antigen, with 12 showing seropositivity for 2 antigens of the same orthoebolavirus, despite not having a prior Ebola disease diagnosis. Furthermore, the majority of respondents reported frequent visits to the forest to hunt a variety of wild animals, particularly ungulates and rodents, which could account for occasional viral spillovers. The absence of active Ebola virus circulation in wildlife may reflect seasonal patterns in reservoir ecology, as those observed in bats. Similarly, seasonal human activities, such as hunting and foraging, may result in periodic exposure risks. These findings highlight the importance of continuous, multidisciplinary surveillance to monitor changes in seasonal spillover risks.

## Author summary

Since its discovery in 1976 in the Democratic Republic of Congo (DRC), Ebola virus (EBOV) has caused more than 20 outbreaks in humans, with fatality rates as high as 90%. While the virus is believed to have an animal origin, naturalist reservoir and the mechanisms of transmission to humans remain poorly understood. Gaining insight into which species may harbour the virus and how transmission occurs is essential to predict and prevent future outbreaks. In this study, we investigate EBOV exposure in wildlife and humans in a region of the DRC with a documented history of outbreaks. Although we did not detect active infection in animals, we found serological evidence of prior exposure in several bat and rodent species, as well as among local residents. Interviews with community members revealed frequent contact with wildlife through hunting and handling, practices that could elevate the risk of animal-to-human transmission. These findings offer new clues about possible EBOV reservoirs and highlight the role of human behaviours in facilitating facilitate spillover events. Our results underscore the need for continued, integrated surveillance to improve understanding of Ebola virus ecology and to help reduce the risk of future Ebola outbreaks in endemic regions.

## Introduction

Ebola disease is a severe viral illness characterised by haemorrhagic fever caused by viruses of the family *Filoviridae*, genus *Orthoebolavirus*. From 1976 to 2023, these viruses have led to 39 confirmed emergences across Africa [1]. The largest outbreak occurred between 2014 and 2016 in West Africa, where it caused over 28 000 reported cases and more than 11 000 deaths [1]. So far, six orthoebolaviruses have been identified: Ebola virus (EBOV, formerly known as Zaire ebolavirus), Sudan virus (SUDV), Bundibugyo virus (BDBV), Taï Forest virus (TAIFV), Reston virus (RESTV), and Bombali virus (BOMV). Among these, EBOV is the most lethal, with a fatality rate

of up to 90% in humans in certain outbreaks [1]. It has also been responsible for most outbreaks, followed by SUDV and BDBV [1].

EBOV was first identified in 1976 in Yambuku, located in the Equateur province of the Democratic Republic of the Congo (DRC). Since then, EBOV has caused a total of 14 confirmed emergences in various regions of the DRC [1]. One occurred in 2014 in the Boende Health District (Tshuapa Province, DRC), lasting from July 26 until October 7. This outbreak resulted in 69 reported cases (suspected, probable, and confirmed) and 49 deaths [2]. Most of these cases were reported in the Djera Sector, a forested area south of Boende. The outbreak's index case was traced back to a pregnant woman from the village of Inkanamongo, who had butchered a monkey found dead by her husband [2].

Most outbreaks caused by EBOV, including the Inkanamongo outbreak, are believed to have a zoonotic origin [2–4]. Nevertheless, despite extensive research efforts, the animal reservoir of the virus remains unidentified [4]. Over 12 000 and 34 000 wild animals have been screened respectively for EBOV RNA or antibodies between 1978 and 2023 to elucidate the reservoir and other animal hosts of EBOV (S1 Table) [4–10]. Gorillas, chimpanzees, and duikers have been suspected (and in the case of chimpanzees, confirmed) to have infected the index case during outbreaks in Gabon and the Republic of Congo between 1996–2003 [11–13]. EBOV RNA has been detected in duikers, and both RNA and antibodies against EBOV have been found in great apes [13,14]. Although the presence of antibodies indicates prior exposure to the virus and detection of EBOV RNA suggests recent infection [11,15], their high disease-related mortality rate and limited distribution make these animals improbable reservoirs of EBOV [14,16].

Substantial evidence suggests that bats, or at least particular species, may serve as a reservoir for EBOV [6,7,9,10,17–21]. Bats have been linked to the start of two different EBOV outbreaks in the DRC (in 2007, the index case had bought freshly killed fruit bats for consumption [17]) and in West Africa (the 2014–2016 outbreak likely originated from a spillover event involving a bat and a two-year-old boy [18]). Supporting this theory, EBOV antibodies were detected in at least eight frugivorous and two insectivorous Old World bat species [6,7,19,20]. Additionally, viral RNA has been found in three of these species (*Epomops franqueti*, *Hypsignathus monstrosus*, and *Myonycteris torquata*) [21]. Unlike apes or duikers, bats have not shown EBOV disease-related mortality, further indicating their potential role as a natural reservoir for the virus. Other filoviruses have also frequently been found in bats: *Rousettus aegyptiacus* is an important reservoir for the zoonotic Marburg virus, and Bombali, Lloviu and Měnglà viruses have so far only been detected in bats (respectively *Mops condylurus*, *Miniopterus schreibersii*, and *Rousettus sp.*) [22–25].

The notorious absence of EBOV detection in wildlife or identification of the reservoir, despite extensive research effort, and in contrast to relatively frequent detections of other filoviruses in wild bats, could be due to several reasons. Possibly, EBOV circulates in a limited number of species which have not been sampled sufficiently up to date. Considering the high mammalian diversity in Afrotropical forests, obtaining a sufficient sample size for each of the hundreds of species is unrealistic.

Furthermore, EBOV infection dynamics could be highly seasonal, following the host reproductive cycles and life phases, and potential migrations, as has been observed for other mammalian-borne infections (e.g., filoviruses, coronaviruses, hantaviruses) [26–31]. The timing of many of the reservoir-searching studies (often conducted a few weeks after the start of an EBOV outbreak in humans) could have coincided with low incidence periods due to these seasonal changes in the natural epidemiology, making the detection probability too low.

Earlier studies have also investigated how socio-cultural factors and human behaviours contribute to the emergence of EVD outbreaks [32]. For instance, the social dimensions of EVD and viral haemorrhagic fevers were studied in Sierra Leone and Guinea [32,33]; bats and wild meat hunting and consumption and perceptions of disease risk were studied in Cameroon [34–36] and Ghana [37]. In the DRC, studies have linked a human EVD outbreak to exposure to fruit bats [17], highlighting that contact with bats, rodents, and eating non-human primate meat was associated with EBOV seropositivity, even in the absence of EVD diagnosis [38]. Additionally, multiple studies have investigated community beliefs about the origins of EVD [35,39,40].

Despite considerable efforts to identify the EBOV reservoir species and spillover pathways, extensive studies that simultaneously cover virological, ecological, sociological, and human health aspects are rare [41]. In line with the "One Health" concept, our study investigates EBOV circulation and health risks associated with wild meat hunting and consumption in a Congolese village (Inkanamongo, Boende Health District) where an Ebola outbreak occurred in 2014. Our study specifically aimed to (i) pinpoint the animal reservoir and ecological factors involved in EBOV maintenance and transmission, (ii) assess past human exposure to EBOV via a serological assay, and (iii) explore interactions between humans, wildlife, and the environment via sociological questionnaires. Although most of the confirmed Ebola disease emergences in the DRC have been caused by EBOV, we did not limit our study to EBOV and also screened for other orthoebolaviruses. This study was conducted in the framework of the EBOVAC3 project, which ran a clinical trial (EBL2007) to assess the safety and immunogenicity of an Ebola vaccine in health care providers in Boende and aims to characterise outbreak preparedness through social science research.

## Materials and methods

### Ethical clearance and permits

Ethical clearance for the human serological study was provided by the Democratic Republic of the Congo's National Committee for Health Ethics (N°529/CNES/BN/PMMF/2023, approved on 02/08/2023). The formal written informed consent was obtained by all participants. Ethical clearance for the sociological studies was approved by the Public Health Ethics Committee of the University of Kinshasa and provided by DRC's National Committee for Health Ethics (N°368/CNES/BN/PMMF/2022, approved on 5/07/2022). The formal verbal informed consent was obtained by all participants. Ethical clearance for the animal reservoir study was provided by the Ethical Committee for Animal Testing of the University of Antwerp (N°2020–22, approved on 03/04/2021). The Nagoya permit for the export of samples from the animal reservoir study was issued by the Ministry of Environment and Sustainable Development, General Secretariat, in Kinshasa (N°009/ANCCB-RDC/SG-EDD/BTB/09/2020 [Ministère de l'Environnement et Développement Durable, Secrétariat Général à l'Environnement et Développement Durable – in French]). The CITES permits for the export out of DRC and import in Belgium of the samples from CITES listed species were provided by CITES/DRC Management Authority in Kinshasa (N°CDFF1918R, CDFF1919R, CDFF1920R, CDFF1921R [Organe de gestion CITES/RDC – in French]) and the FPS Health, Food Chain Safety and Environment, DG Environment, Multilateral Affairs and Strategic Matters, CITES Management Body, Brussels, Belgium (N°2022/BE00308/PI [FOD Volksgezondheid, Veiligheid Voedselketen en Leefmilieu, DG Leefmilieu, Dienst Multilaterale en Strategische Zaken, Beheersorgaan CITES – in Dutch]). All "movement of personnel" (called "Ordre de Mission") were signed and approved by the local authorities at each site. Material transfer agreements (MTA) were issued by the University of Kisangani, Biodiversity Surveillance Center (CSB [Centre de Surveillance de la Biodiversité] – in French). Live captured animals were euthanised (isoflurane) following the 2013 AVMA Guidelines for the Euthanasia of Animals and Sikes and Gannon 2007 (J Mammal. 88:809–23).

### Animal reservoir study

**Trapping and sampling of wild mammals.** Considering the assumed seasonality of EBOV outbreaks in the DRC, we conducted the ecological survey from May 13 to June 11, 2021, which coincides with the months preceding the 2014 outbreak (June-July). We trapped 633 small mammals (rodents, shrews, and bats) using a variety of traps within a 3 km radius of Inkanamongo (S1 Fig). Additionally, we sampled 247 mammals provided by local inhabitants, including rodents captured in and around their homes and larger wildlife caught by hunters deeper in the forest. On top of that, we also sampled 34 livestock individuals. We refer to the supporting information for more details on the sampling process (S1 Supporting Information).

We dissected all captured small terrestrial mammals, but bats were kept alive and only dissected in certain cases listed in the supporting information (S1 Supporting Information). Live-captured terrestrial animals were euthanised by an

overdose of isoflurane before the collection of blood on Whatman filter paper (hereafter called dried blood spots (DBS)) and tissue samples in DNA/RNA shield or 99% ethanol. For bats, we collected oral and rectal swabs, faeces, and urine (if available) stored in DNA/RNA Shield. We also performed a 3mm diameter wing punch biopsy for molecular identification and stored it in 99% ethanol. Whole blood was collected as DBS by puncturing the cephalic vein.

**Molecular work.** The inactivated samples in DNA/RNA shield were transported to the University of Antwerp and processed under a Biosafety cabinet class II for molecular species identification (via sequencing of the cytochrome b gene) and filovirus RNA detection. The latter was carried out using two previously published PCR assays with degenerate primers targeting a fragment of the filovirus polymerase (L) gene [42,43]. Full details of the procedures are provided in the supporting information (S1 Supporting Information) and in S2 Table.

**Animal serology.** We used a multiplex serological assay (Luminex-based technology) to test DBS from bats and rodents for anti-orthoebolavirus IgG antibodies, as previously described [6,9,10]. In brief, the assay incorporated recombinant orthoebolavirus proteins, including glycoprotein (GP), nucleoprotein (NP), and viral protein 40 (VP40) from five different orthoebolaviruses (EBOV, SUDV, BDBV, RESTV, and BOMV). To identify a sample as positive, we used various combinations of antigens for which the median fluorescence intensity (MFI) exceeded the estimated cutoffs. In the absence of established wildlife control samples, we determined cut-off values using several methods tailored to the available data. For bats, we calculated the mean MFI cut-off values based on four different approaches. This included the mean MFI plus 4 times the standard deviation from 145 negative samples obtained from bats housed in European zoos [6], as well as the 97.5th percentile cut-off derived from 3 statistical analyses (change point, exponential, and binomial) using a larger dataset of wild bat samples (totalling 8741 individuals from Guinea, DRC, and Cameroon) [9]. We considered a sample likely to be positive when it reacted to more than one antigen of the same orthoebolavirus above the cutoff values. All details are provided in the supporting information (S1 Supporting Information).

This assay has been extensively validated in prior bat studies by our laboratory [6,9,10], demonstrating reproducibility and robust statistical validation. We selected the Luminex platform over Western blotting or neutralization assays because of its high sensitivity, low sample volume requirements, and compatibility with dried blood spots, which also offer major biosafety and logistical advantages. The same methodology has been applied for EBOV antibody detection in human sera where it showed specificity and sensitivity comparable to or higher than commercial ELISAs [44], supporting its broader applicability for serological analyses in wildlife.

For the rodent samples, due to the lack of validated negative control samples and a comprehensive panel of previously tested wildlife specimens, we calculated cut-offs directly from our study samples. Specifically, the MFI cut-off was set at the 97.5th percentile, recognising that this method inherently produces at least a 2.5% false positive rate. To mitigate this, we only classified a sample as positive if both the nucleocapsid and glycoprotein of the same orthoebolavirus exceeded their respective cut-offs. Because both antigens are not correlated in false-positive samples, this criterion significantly reduces the chances of false-positive results. We performed the analyses with R version 4.3.0 software [45].

## Human serology study

A cross-sectional study was conducted in Inkanamongo in the Boende Health District and the Lokolia Health Area where there was an Ebola virus disease (EVD) outbreak in 2014. The door-to-door survey was conducted from December 2–5, 2023, to estimate seroprevalence of orthoebolaviruses in the human population with a sample size of 380 adults who had never received a vaccine against EBOV. Blood samples were collected on Whatman 903 filter paper from each participant's fingertips. Each participant provided information on socio-demographic characteristics, knowledge of and information channels about EVD, whether they had experienced symptoms of the disease, and whether they or a close relative had contracted EVD in the past. Samples were tested for anti-orthoebolavirus antibodies using the same Luminex assay as previously described for wild animals, but with positivity-cutoffs calculated on human positive and negative control samples as explained in detail in [46]. To identify participant characteristics associated with past EBOV infection,

we developed different (Generalized) Linear Models for each EBOV antigen with IgG titres (expressed as the logarithm of MFI values) or antibody presence (higher or lower than cutoff) as response variable and age, sex, and previous EVD (no, yes, or indirect contact only via family member) as explanatory variables [47]. We conducted our analyses using R version 4.2.2 software [45]. The significance of the explanatory variables was evaluated using χ2 tests (p-values) or by assessing changes in the Akaike Information Criterion (ΔAIC), which involved comparing the full model against models with individual variables removed.

### Social science study

We conducted two types of social science studies: interview sessions in Inkanamongo (referred to as Study A) and a more structured study conducted in villages of the entire Djera Sector (referred to as Study B). **Study A** corresponds to observations and interview sessions conducted in Inkanamongo between May 17 and May 27, 2021, simultaneously with the animal samplings. This study consisted of two sub-surveys. During the first one, observation notes were taken about the inhabitants' daily activities. Following a convenience sampling method, 40 inhabitants were interviewed with the help of a structured questionnaire to assess (i) their daily activities and (ii) their behaviours regarding animal capture, specifically assessing local hunting behaviours and preferences. The second sub-survey focused specifically on residents' (i) awareness, perceptions, and knowledge of bats, (ii) hunting habits regarding bats and (iii) beliefs or practices associated with bats. Ten persons were interviewed with the help of a structured questionnaire. Additional information on study A and the structured questionnaires are provided in the supporting information (S1, S2 and S3 Supporting Information).

   **Study B** refers to a dedicated survey mission conducted in six villages of the Djera Sector most affected by Ebola in 2014 (Inkanamongo, Lokolia, Ituku, Watsi Kengo, Isaka, Iyute; S1 Fig). Here, we interviewed 117 inhabitants of these villages to understand the potential of animal-to-human and human-to-human transmission of EBOV by assessing (i) the people's interactions with the forest by investigating their use and knowledge of the environment (agriculture, fishing, and hunting); and (ii) their interactions between and within human communities. Data was collected door-to-door between July 16 and July 23, 2022. All details and the structured questionnaire are provided in the supporting information (S1 and S4 Supporting Information) and S3 Table.

## Results

### Animal reservoir study

We collected samples from 922 animals belonging to 60 different species within eight orders (Table 1). Details on the molecular species identification and screening results of all sampled specimens are provided in S4 Table. Overall, we did not detect any active EBOV infection in 1049 screened samples (464 pooled non-invasive samples and 585 pooled kidney and liver) from 919 individuals (Table 1).

   Among the 296 bats screened, 13 bats from 4 species were reactive with at least 1 orthoebolavirus antigen (Tables 2 and S4). All bats that showed reactivity belonged to the Pteropodidae family: *H. monstrosus* (2/13), *E. franqueti* (3/13), *Eidolon helvum* (2/13), and *M. torquata* (6/13). Cross-reactivity among the same antigens of different orthoebolaviruses was observed, with two bats reactive with the GP antigens of EBOV, SUDV, and BDBV, and one bat reactive with the VP40 antigens of EBOV, SUDV, and BDBV. However, no bats were reactive with combinations of different antigens (e.g., NP + GP or GP + VP40) from the same orthoebolavirus.

   From the 268 rodents tested, 38 rodents from 11 different species were reactive with at least 1 orthoebolavirus antigen (Tables 2 and S4): *Congomys lukolelae* (1/38), *Funisciurus anerythrus* (2/38), *Hylomyscus* cf. *aeta* (1/38), *Lophuromys rita* (15/38), *Mus* cf. *gratus* (3/38), *Oenomys hypoxanthus* (1/38), *Praomys minor* (2/38), *Praomys jacksoni* complex sp. 'IVb' (1/38), *Rattus rattus* (9/38), *Stochomys longicaudatus* (1/38) and *Thamnomys poensis* (2/38). However, given that the changepoint analyses suggested a considerably higher number of positives than the mean+4SD method (37 vs. 17), we

**Table 1. The number of animals sampled, tested with PCR, and tested with Luminex per taxonomic order.**

| Order | Field-trapped | Provided by hunters/villagers | Livestock in village | Tested with PCR | Tested with Luminex |
|---|---|---|---|---|---|
| | | | | N tested* | Pos/N tested (%) |
| Carnivora | 0 | 2 | 0 | 2 | 0 |
| Cetartiodactyla | 0 | 4 | 34 | 38 | 0 |
| Chiroptera | 312 | 70 | 0 | 381 | 13/296 (4.4) |
| Eulipotyphla | 130 | 1 | 0 | 130 | 0 |
| Macroscelidea | 0 | 41 | 0 | 41 | 0 |
| Pholidota | 0 | 3 | 0 | 3 | 0 |
| Primates | 0 | 19 | 0 | 19 | 0 |
| Rodentia | 199 | 107 | 0 | 305 | 38/268 (14.2) |
| Total | 641 | 247 | 34 | 919 | 51/564 (9.0) |

The number of animals reactive with at least one orthoebolavirus antigen and the percentage are presented.

* There were no PCR positives.

**Table 2. Number of individuals that tested positive against different orthoebolavirus antigens using a Luminex assay.**

| Taxon | Antigen | EBOV | | | | SUDV | | | BDBV | | RESTV | BOMV |
|---|---|---|---|---|---|---|---|---|---|---|---|---|
| | | NP | GP-K | GP-M | VP40 | NP | GP | VP40 | GP | VP40 | GP | GP |
| **Chiroptera N = 296** | Mean+ 4SD¹ | 0 | 2 (0.7) | 1 (0.3) | 1 (0.3) | 2 (0.7) | 7 (2.4) | 3 (1.0) | 3 (1.0) | 1 (0.3) | 1 (0.3) | – |
| | Mean 3 cutoffs² | 0 | 1 (0.3) | 1 (0.3) | 1 (0.3) | 0 | 1 (0.3) | 2 (0.7) | 0 | 1 (0.3) | 1 (0.3) | – |
| **Rodentia N = 268** | Mean+ 4SD³ | 3 (1.1) | 5 (1.9) | 5 (1.9) | 1 (0.4) | 1 (0.4) | 5 (1.9) | 1 (0.4) | 5 (1.9) | 2 (0.7) | 3 (1.1) | – |
| | Mean 3 cutoffs³ | 7 (2.6) | 16 (6.0) | 19 (7.1) | 2 (0.7) | 4 (1.5) | 18 (6.7) | 1 (0.4) | 5 (1.9) | 1 (0.4) | 3 (1.1) | – |
| **Humans N = 380** | Mean 3 cutoffs⁴ | 8 (2.1) | 23 (6.1) | 42 (11.1) | 19 (5.0) | 14 (3.7) | 84 (22.1) | 2 (0.5) | 10 (2.6) | 14 (3.7) | 2 (0.5) | 0 |

Number (%) of positive bats, rodents and humans, based on two different cut-off values (for animals). The assay used recombinant proteins of Nucleoprotein (NP), Glycoprotein (GP), or Viral Protein-40 (VP40) for different orthoebolaviruses: Ebola virus (EBOV), Sudan virus (SUDV), Bundibugyo virus (BDBV), Reston virus (RESTV), and Bombali virus (BOMV). GP proteins from the Mayinga (GP-M) and the Kissidougou (GP-K) strain were used for EBOV.

¹ calculated on a panel of 145 negative samples (bats) [6].

² calculated on a panel of 8741 wildlife samples (bats) [9].

³ calculated on 268 samples from this study (rodents).

⁴ calculated on a panel of 202 human samples [46].

assume that cutoffs might be too low and many samples could be false positives. Nevertheless, three samples are more likely to be true positives given that two different antigens from the same orthoebolavirus scored positive: one *L. rita* tested positive for a combination of VP40 + GP EBOV antigens, another *L. rita* for NP + GP EBOV antigens, and a third *L. rita* showed reactivity against the NP and GP antigens of SUDV. Half of the rodents that showed reactivity against at least one orthoebolavirus antigen were captured in or around the houses (19/38); one of these was the *L. rita* showing simultaneous reactivity for NP and GP EBOV antigens.

## Human serology study

We screened DBS from 380 participants between 18 and 76 years old and found IgG antibodies against at least 1 orthoebolavirus antigen in 120 participants (31.6%, 95% CI: 26.9-36.5%. Fig 1 and Tables 2 and S5). Reactivity to at least 2 antigens of the same orthoebolavirus (much stronger indication for true positive samples) was observed for 12 of 380 individuals (3.2%, 95% CI: 1.6-5.5%), of which 7 reacted to EBOV antigens, 3 to SUDV, 1 to BDBV, and 1 to both EBOV

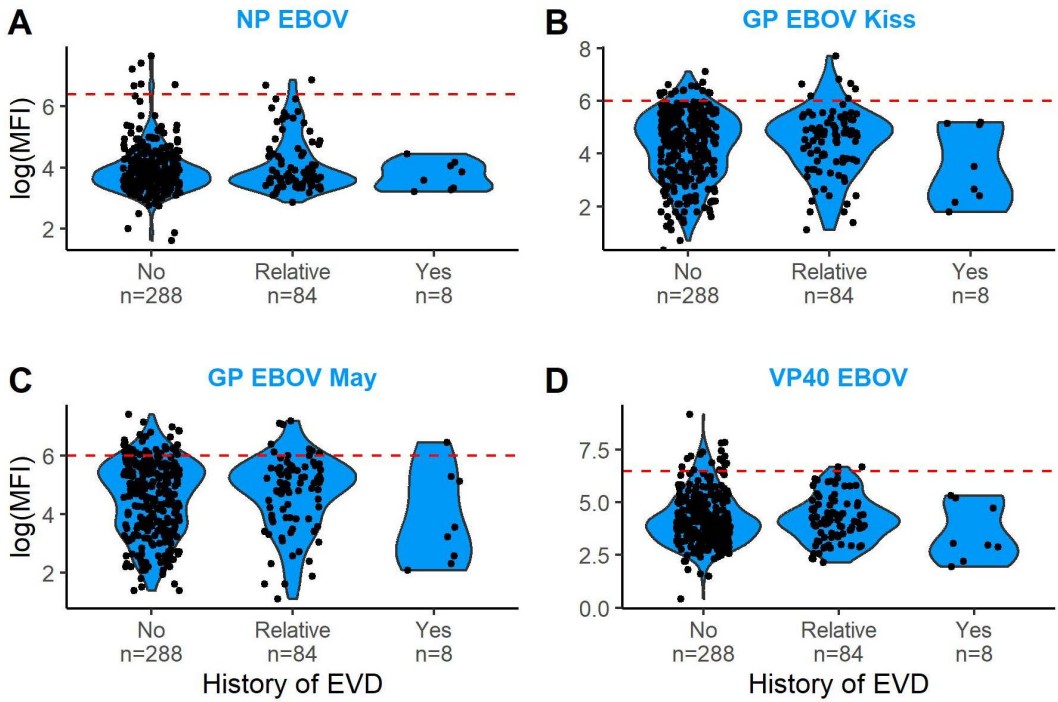

**Fig 1. IgG antibody titres against Ebola virus from inhabitants in Inkanamongo (DR Congo).** Titres are expressed as Median Fluorescent Intensity values (MFI) on the Luminex for four different antigens: nucleocapsid protein (NP), the glycoproteins Kissidougou (GP-Kiss) and Mayaro (GP-May) and the viral protein40 (VP40). Violin plots express the log(MFI) values for three different categories of participants according to previous Ebola viral disease (EVD) exposure: no infection, relative became infected, and participant became infected. The red dotted line indicates the cutoff of the antigen on the Luminex. Positivity cutoffs calculated on a panel of 202 human positive and negative control samples as explained in detail in [46].

and BDBV (S6 Table and S2 Fig). Of these 12 seropositive individuals, 10 were male and 2 were female, and their ages ranged from 18 to 52 years. Interestingly, none of these 12 individuals were previously diagnosed with EVD, although 2 of them reported they had someone close to them who suffered from EVD (S7 Table). In contrast, out of eight individuals who had contracted EVD in the past, only two were found to be reactive with one antigen (GP EBOV and GP SUDV), and none showed reactivity against two antigens. The (generalised) linear models did not suggest any significant effects of age, sex or previous EVD status on the presence or absence of antibodies or their titres against EBOV (S8 Table).

## Social science study

Two surveys in Inkanamongo and nearby villages revealed strong reliance on forest-based livelihoods and diverse wildlife use. Table 3 summarises key findings on livelihood strategies, environmental perceptions, wildlife interactions, and health behaviours.

When combining data from the two surveys (A and B), Cetartiodactyla and Rodentia represented the orders most frequently reported as observed (Fig 2). Molecular identification of sampled animals hunted for consumption collected in Inkanamongo confirmed the presence of most of the animals mentioned by hunters in the sociological study. Overall, we identified at least 9 orders, 25 genera, and 27 species of animals hunted around Inkanamongo (S9 Table). Notably, while bats (Chiroptera order) were not reported as hunted in the broader A and B surveys, 1 out of 10 respondents in the bat-specific investigation reported hunting them and one reported consuming them. Similarly, we observed people in the village with dead elephant shrews (Macroscelidea order), but none of the respondents reported having observed or hunted them. Last, gorillas, chimpanzees, and lions were reported as observed by respondents, although they do not

**Table 3. Key findings from sociological study.**

| Theme | Key finding | Main figures | Study (N) |
|---|---|---|---|
| Livelihood | • Multi-activity (hunting, fishing, farming, etc.)<br>• Strong reliance on forest-based resources<br>• Hunting mostly secondary or tertiary activity | 100% → agriculture<br>80% → hunting<br>80% had > 1 occupation | A (N = 10) bat-specific |
| | | 62.5% → agriculture<br>80% → hunting<br>70% → fishing<br>77.5% had > 1 occupation | A (N = 40) |
| | | 97% → agriculture<br>50% → hunting<br>57% → fishing<br>91.4% had > 1 occupation | B (N = 117) |
| Forest access | • Frequent interaction with forest environment | 46% → daily<br>32% → 2–3 times/ week | B (n = 117) |
| Seasonality of resources use | • Seasonal variation in hunting and harvesting | 26/40 prefer wet season; 4/40 dry season; 10/40 no preference; bats used as fruit ripening indicators | A (N = 40 + 10) General+ bat-specific |
| Forest animal diversity | • Wide range of species observed in the forest | 13 taxonomic orders observed | A + B (N = 129) |
| Bushmeat and fish trade | • Local and regional trade, with dominance of local trade | 100% report local consumption; 42.5% mention trade to Boende | A (N = 40) |
| | | 65% sell; 68.4% buy; mostly local trade; some trade with Boende | B (N = 117) |
| Wild meat consumption | • Broad consumption | 68.4% buy wild meat; 93% consume wild meat | B (N = 117) |
| Wild meat animal diversity | • Variety of species hunted and consumed<br>• Partial confirmation of sociological data with molecular data | 9 orders hunted:<br>34.5% Cetartiodactyla (antelopes, sitatunga, bushpig)<br>29.8% Rodentia (porcupines, squirrels, rats)<br>28 species identified (molecular) | A + B (N = 129) |
| Cultural beliefs | • Ethno-cultural perceptions and symbolic meanings of bats<br>• Cultural beliefs shape bat classification, hunting roles, and consumption taboos | Interviews with 10 adults from 6 Mongo clans (Ekolie, Djwangenda, Djiyoko, Nongo, Djeefo, Djosona); Ekolie hunt/sell but don't consume bats; 3 Ekolie respondents reported that bat vocalizations signal misfortune | A (N = 10) bat-specific |

occur in the area. We provide in the supporting information (S10 Table) a comprehensive list of animals reported as present in the study area, as well as those for which a translation of the local names used during the interviews was available in the literature.

## Discussion

The detection of EBOV in wildlife remains challenging, as highlighted by the limited number of studies and the few species in which the virus has been detected [11,15,21]. Notably, all studies conducted in the DRC, both during and after EBOV outbreak periods, including ours, have been unsuccessful in detecting EBOV RNA in wildlife. A key factor contributing to this challenge may be the underrepresentation of certain species in sampling efforts, which can distort our understanding of the virus' true host range and prevalence. For instance, species that are difficult to access due to their habitat, behaviour, or rarity may serve as undetected reservoirs or be critical to overlooked transmission pathways. Addressing this issue requires the development and implementation of systematic and balanced sampling strategies across a broad spectrum of species to accurately identify reservoirs and better understand interspecies transmission.

Although we did not detect any EBOV RNA in wildlife around Inkanamongo, we identified antibodies against at least one EBOV antigen in some animals, suggesting past exposure to the virus or related orthoebolaviruses. Using the same methodology as Lacroix et al. [9], we found seroprevalence in our bats to range from 1.2% to 4.0% in Inkanamongo.

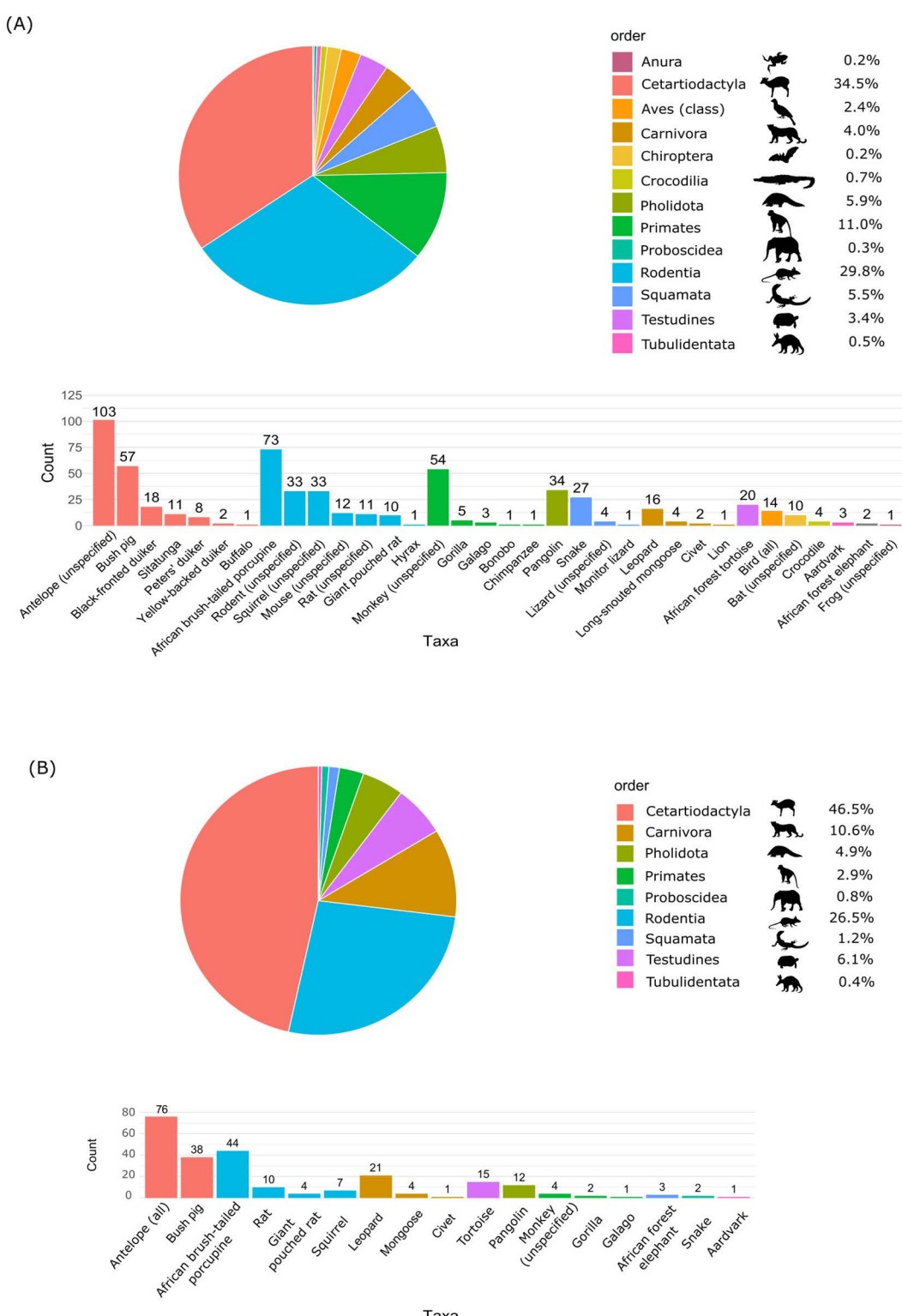

**Fig 2. Mentions of different vertebrate taxa reported as seen in the forest and mentioned as hunted.** (A) seen in the forest, N = 580 occurrences, all villages including Inkanamongo. (B) hunted, N = 245 occurrences, all villages including Inkanamongo. Thirty mentions could not be assigned to any taxa or order as no translation could be found. We provide in the supporting information a table with the translation of the different taxa when available (S10 Table).

Similar to our findings, Lacroix et al.'s [9] samples generally did not show simultaneous reactivity to different EBOV antigens, except for one *Epomophorus* bat from Nord Kivu province. We observed lower reactivity to NP compared to GP and VP40 in bats, and no NP cross-reactivity. Since both NP and VP40 were produced in *Escherichia coli*, this difference likely reflects antigen-specific immunogenicity or antibody kinetics rather than expression system. Some studies report faster waning of NP-specific antibodies [48], while others found NP responses persisting decades post-infection [49], suggesting variability across contexts. Our two-antigen criterion for positivity increases specificity but may miss past infections if one response, such as to NP, has declined. We therefore interpret single-antigen reactivity with caution. While the overall seroprevalence in our bat samples appears lower compared to some studies conducted in the DRC and other endemic regions [7,9,10], direct comparisons are challenging. For example, Lacroix et al. [9] reported seroprevalence rates ranging from 1.2% to 10.8% in bats, depending on the positivity criteria. However, their data combined samples from multiple species across different sites and seasons, with uneven sample size per species, making direct comparisons difficult. For example, the seroprevalence in *E. helvum*, *E. franqueti*, *H. monstrosus*, and *M. torquata* varied across regions of the DRC and sampling seasons which underscores how ecological factors such as site-specific conditions and seasonality can significantly influence EBOV antibody detection. Another serological survey conducted in 2023 across Western and Central Africa using the same methodology found even higher seroprevalence estimates in bats, ranging from 6.1% to 18.9% from more than 15 species collected between November 2015 and September 2020 over 31 sites [10]. Ultimately, these discrepancies highlight the challenges of comparing seroprevalence data due to varying ecological settings, sampling dates, and seasons, all of which can influence the observed results.

Age, reproductive status, and seasonal behaviours of forest animals, including bats and terrestrial mammals, may significantly influence the circulation of EBOV and observed seroprevalence patterns. Younger animals, often immunologically naïve, may amplify viral transmission within populations, particularly during birthing seasons when juveniles are abundant and highly susceptible [50]. Reproductive status may be another key factor, as pregnant or lactating individuals may experience altered immune responses due to hormonal changes, nutritional stress, and increased energy demands, potentially enhancing susceptibility to infection or extending viral shedding [50]. These changes can elevate susceptibility to infection or prolong viral shedding, although findings on EBOV remain mixed. For instance, while higher seroprevalence was observed in gestating or lactating *E. helvum* and *Micropteropus pusillus*, other species like *H. monstrosus* showed lower seroprevalence, highlighting the variability across species and contexts [10,51]. Seasonal migration and foraging behaviours further complicate this dynamic, with the movements of animals potentially introducing the virus to new areas or intensifying local outbreaks [52]. Consequently, the seasonal migration of bats in large numbers has been hypothesised as another important risk factor in previous outbreaks [17]. Ecological surveys conducted in Occidental Kasaï (DRC) suggested that the large seasonal migrations of fruit bats between April to May may have started the 2007 outbreak in Kasai. During this period, bats (including *H. monstrosus*, *E. franqueti*) were likely hunted in large numbers by villagers, corresponding to the observation that the putative index case obtained freshly killed bats from hunters. Even though no data can be found about the migratory behaviour of *H. monstrosus* and *E. franqueti*, hunters from these villages reported massive annual migrations of these species in the area [17]. Similarly, during the Likati outbreak in 2017, a large seasonal colony of *E. helvum* bats was roosting at the Likati River from March to July [4]. The human index case bought and consumed a large bat during this time. During our reservoir study in Inkanamongo, we witnessed the arrival of *E. helvum* bats from May to June. Notably, while not a single specimen of this species was observed during the initial two weeks of our sampling in May 2021, we started to capture some in our nets in week three and noticed hunters returning to the village with them. Combined, these biological and ecological factors create fluctuating eco-epidemiological conditions that may influence EBOV transmission and seroprevalence, underscoring the need for targeted, seasonally informed surveillance strategies.

The interpretation of our serological findings in rodents requires caution due to the absence of validated negative control samples, which can influence the determination of a biased cut-off value. Nonetheless, three rodents tested positive

for two antigens of the same orthoebolavirus, meeting a more stringent serological criterion [6,9]. Interestingly, all were *L. rita*, of which one was captured in the village. Additionally, traces of EBOV RNA were previously detected in three rodent taxa (*Mus setulosus* and two unidentified *Praomys* species) and one shrew (*Sylvisorex ollula*), although antigen detection and attempts to isolate the virus failed [53]. Furthermore, several studies have shown that filovirus-like elements are integrated into the genomes of certain rodents, shrews, and bats, indicating a long-term history of mammal-filovirus association [54,55]. Modelling has also suggested bats, rodents, and shrews as primary candidates for EBOV reservoirs [56,57]. However, as conclusive empirical evidence is still lacking, increased surveillance, accounting for potential seasonal variations, is needed to elucidate the role of rodents and shrews in EBOV circulation.

Another objective of our study was to assess human exposure to EBOV in Inkanamongo. While almost one-third of the inhabitants (n = 120) had IgG antibodies against at least 1 orthoebolavirus antigen, we identified only 12 seropositive participants who were positive for at least 2 antigens of the same orthoebolavirus. Notably, one participant was found to be seropositive for both EBOV and BDBV. Given that the only known outbreak of BDBV in the DRC happened over 800km from our study site, this observation rather suggests cross-reactivity between orthoebolaviruses. Surprisingly, none of these 12 seropositive participants had a history of EVD. This finding is consistent with several other studies in which they detected EBOV antibodies in individuals with previously undetected EBOV infections and no EVD-related symptoms [5,33,47,58]. Although two-thirds of our participants reported experiencing EVD-like symptoms such as fever and vomiting in the past months, these symptoms could also be attributed to other diseases, and we cannot exclude the possibility of false-positive results. Only 14 participants reported bleeding episodes with one person reporting internal bleeding and 13 external bleeding. Nevertheless, our results indicate that the inhabitants of Inkanamongo are occasionally exposed to orthoebolaviruses in their surroundings and/or that EBOV may present as a minimally symptomatic or asymptomatic infection. Additionally, we observed that none of the previously EVD-infected survivors in the villages had antibodies against at least two EBOV antigens. This indicates that humoral immunity may decline more rapidly than anticipated, a phenomenon also observed in survivors of the 2013–2016 EBOV outbreak in Sierra Leone [59] and Guinea [48] but contrasts with findings from Rimoin et al. [49], which reported the persistence of EBOV antibodies 40 years postinfection.

Finally, we investigated the interactions between humans and wildlife in the villages. Our findings indicate that over 50% of the humans in the area frequently visit the forest to hunt various wild animals, particularly ungulates and rodents. However, we noticed inconsistencies between the respondents' reported behaviour and our field observations concerning human-wildlife interactions. We observed inhabitants with dead elephant shrews although they did not report hunting them, they may have misclassified them as rodents. Additionally, although only one respondent mentioned hunting bats in the region, we documented instances where some people engaged in selling, hunting, or consuming bats in Inkanamongo. These results highlight the reliance of local populations on wild meat, which is widely acknowledged by respondents: over 90% reported consuming wild meat, approximately two-thirds reported selling it and a similar proportion reported buying it, primarily locally but also at the regional level. Yet, the discrepancy between interviews and field observations, especially regarding bats, suggests selective underreporting shaped by stigma, cultural norms or awareness of health risks in the post Ebola context. Also, specific interviews on bats revealed that ethnic-based perceptions and tradition are important factors to consider when investigating interactions with bats. Such sociocultural aspects have been partially investigated in the context of EVD emergence [60]. However, epidemiologic links between wildlife consumption or various aspects of practices and infections remain poorly understood. Disentangling the health risk factors and socioeconomic dynamics of disease emergence remains a complex issue and requires further interdisciplinary investigation in the future [61].

Furthermore, during our questionnaire administrations, significant uncertainties emerged about the origin of the 2014 Ebola outbreak in Inkanamongo. While Maganga et al. [2] reported that the outbreak started on the 26th of July 2014 in Inkanamongo, the head nurse at Lokolia Health Centre (a pivotal figure in the local health system) stated that the index case originated in Tomoge, a village a few kilometres from Lokolia. He mentioned that the index in this village (the wife of a

teacher) discovered a dead antelope (*Tragelaphus eurycerus* or bongo) in the forest and consumed it with her family. This index case became ill on July 15th, 9 days preceding the documented start of the outbreak by Maganga et al. [2]. Interestingly, the individual considered to be the official index case from Inkanamongo was the 28th person to be recorded with Ebola-like symptoms, according to the Lokolia Health Centre records. According to the head nurse, this person fell ill after visiting an infected woman in Inkanamongo, who had contracted it from another village (personal correspondence). This alternative version of the outbreak is coherent with the testimony of the husband of the presumed index case that we interviewed in Inkanamongo. This questioning of the outbreak narrative, years after the initial investigation into the outbreak's origin, highlights the challenges of epidemiological traceability. Accurate identification of the index is crucial for eco-epidemiological studies and subsequent field missions aimed at monitoring host animals and zoonotic diseases. Misidentifying the index case may lead to incorrect geographical areas and potentially to the wrong suspected reservoir animals of the outbreak.

## Conclusion

Despite not detecting active EBOV infection in our samples, our serological results indicate that inhabitants and wildlife in this region are occasionally exposed to EBOV or other orthoebolaviruses. These findings align with our sociological study, which revealed that most people frequently enter the forest to hunt various wildlife that are suggested to be potential EBOV reservoirs, including rodents, bats, antelopes, and primates. We hypothesise that the lack of active EBOV infection during our study may be due to low sample size for a high number of species sampled, or because of seasonal patterns in the migration or reproductive cycles of the presumed reservoir host. Additionally, we noticed that changes in human activities, such as increased hunting and foraging during periods of food abundance (e.g., caterpillars, fruits, animals), could periodically elevate the risk of wildlife disease spillover. Therefore, we recommend longitudinal surveillance of people and animals to monitor seasonal variations in the presence of EBOV or other zoonotic diseases in endemic villages.

## Supporting information

**S1 Table.  Overview of previous studies investigating Ebola virus in wild animals from 1976 to 2023.**
(XLSX)

**S2 Table.  Primers used in this study.**
(XLSX)

**S3 Table.  Number of persons interviewed in every village in Djera Sector.**
(XLSX)

**S4 Table.  Details of all animals sampled, screened by PCR and screened with Luminex per order and species.**
Species of which certain individual(s) reacted with antigens of orthoebolaviruses are marked in red. Species identification was done molecularly.
(XLSX)

**S5 Table.  Age, sex, EVD history and MFI values for the different orthoebolavirus antigens in humans.**
(XLSX)

**S6 Table.  Number and percentage of samples reactive with at least two antigens of the same orthoebolavirus out of 380 total samples from humans.**
(XLSX)

**S7 Table.  Reactivity to zero, one or two antigens of the same orthoebolavirus depending on the history of contact with Ebola.**
(XLSX)

**S8 Table. Overview of the different models used to test the effect of EVD, age, and sex on MFI and seropositivity in humans for each antigen.**
(XLSX)

**S9 Table. List of animals identified as wild meat from Inkanamongo hunters during the ecological study in 2021.**
(XLSX)

**S10 Table. Animal taxa reported as occurring in or around the Salonga National Park.** Their translation in local language (Lingala or Kimongo), their common name in French, in English, and their Latin species name are given when known.
(XLSX)

**S1 Supporting Information. Context and methods.**
(DOCX)

**S2 Supporting Information. Structured questionnaire of Study A – sub survey 1.**
(DOCX)

**S3 Supporting Information. Structured questionnaire of Study A – sub survey 2.**
(DOCX)

**S4 Supporting Information. Structured questionnaire of Study B.**
(PDF)

**S1 Fig. Sampling area, interviewed villages, and trapping locations.** The interviewed villages are Inkanamongo, Ituku, Lokolia, Iyute, Isaka, and Watsi Kengo. Trapping took place in and around Inkanamongo. Base layers include: Sentinel-2 cloud-free satellite imagery from EOX::Maps Sentinel-2 Cloudless (EOX IT Services GmbH; https://s2maps.eu); administrative boundaries from the Common Geographic Reference Framework (ITOS, 2019) via HDX (https://data.humdata.org/dataset/cod-ab-cod) licensed under CC BY-IGO; physical features and country outlines from Natural Earth (public domain; https://www.naturalearthdata.com/downloads/); hydrological network from OpenStreetMap and OpenStreetMap Foundation via HDX (https://data.humdata.org/dataset/hydrographie-lineaire-rdc-drc-water-courses) under CC BY license; protected areas from the World Database on Protected Areas (2016) via OpenAfrica (https://bulk.openafrica.net) licensed under CC BY 4.0; roads and paths were digitised as vector features, resulting in an original geospatial dataset. All spatial data were used under open licenses and assembled using QGIS.
(TIF)

**S2 Fig. Comparative analysis of IgG responses to Ebola virus antigen combinations in Inkanamongo (DR Congo).** IgG antibody titres in inhabitants of Inkanamongo (DR Congo), expressed as the log of Median Fluorescent Intensity (MFI) values measured by Luminex for four Ebola virus antigens: nucleocapsid protein (NP), glycoprotein from the Kissidougou strain (GP-Kiss), glycoprotein from the Mayinga strain (GP-May), and viral protein 40 (VP40). The data are presented as follows: GP-Kiss and NP (top left), GP-May and NP (top right), VP40 and NP (bottom left), and VP40 and GP-Kiss (bottom right). Each dot represents an individual participant. Participants are classified by Ebola virus disease (EVD) exposure: no infection (black), infected relative (red), and participant became infected (green). The red dotted line represents the antigen-specific cutoff on the Luminex assay.
(TIF)

## Acknowledgments

We wish to thank Josef Bryja and Loic Adrien Mbong Osseke for their contributions to the molecular identification of potential reservoir species and we thank Alisa Victor and Lard Bielen for their contributions to the livestock screening. We also thank the EBOVAC3 logistical team and inhabitants of Inkanamongo for their help during the ecological study.

## Author contributions

**Conceptualization:** Martine Peeters, Séverine Thys, Pierre Van Damme, Jean-Pierre Van Geertruyden, Hypolite Muhindo-Mavoko, Patrick Mitashi, Herwig Leirs, Erik Verheyen, Guy-Crispin Gembu, Joachim Mariën.

**Formal analysis:** Rianne van Vredendaal, Léa Joffrin, Antea Paviotti, Joachim Mariën.

**Funding acquisition:** Sophie Gryseels, Pierre Van Damme, Jean-Pierre Van Geertruyden, Herwig Leirs, Erik Verheyen, Joachim Mariën.

**Investigation:** Rianne van Vredendaal, Léa Joffrin, Antea Paviotti, Claude Mande, Solange Milolo, Nicolas Laurent, Léa Fourchault, Douglas Adroabadrio, Pascal Baelo, Steve Ngoy, Papy Ansobi, Casimir Nebesse, Ahidjo Ayouba, Maeliss Champagne, Julie Bouillin, Jana Těšíková, Natalie Van Houtte, Sophie Gryseels, Maha Salloum, Freddy Bikioli, Séverine Thys, Jimmy Mpato, Ruben Ilonga, Henri Kimina, Hypolite Muhindo-Mavoko, Patrick Mitashi, Herwig Leirs, Erik Verheyen, Guy-Crispin Gembu, Joachim Mariën.

**Methodology:** Rianne van Vredendaal, Léa Joffrin, Antea Paviotti, Nicolas Laurent, Martine Peeters, Ahidjo Ayouba, Sophie Gryseels, Séverine Thys, Ynke Larivière, Gwen Lemey, Pierre Van Damme, Jean-Pierre Van Geertruyden, Hypolite Muhindo-Mavoko, Patrick Mitashi, Herwig Leirs, Erik Verheyen, Joachim Mariën.

**Project administration:** Antea Paviotti, Martine Peeters, Ahidjo Ayouba, Sophie Gryseels, Séverine Thys, Ynke Larivière, Gwen Lemey, Pierre Van Damme, Jean-Pierre Van Geertruyden, Hypolite Muhindo-Mavoko, Patrick Mitashi, Herwig Leirs, Erik Verheyen, Guy-Crispin Gembu, Joachim Mariën.

**Supervision:** Antea Paviotti, Martine Peeters, Ahidjo Ayouba, Sophie Gryseels, Séverine Thys, Ynke Larivière, Gwen Lemey, Pierre Van Damme, Jean-Pierre Van Geertruyden, Hypolite Muhindo-Mavoko, Patrick Mitashi, Herwig Leirs, Erik Verheyen, Guy-Crispin Gembu, Joachim Mariën.

**Visualization:** Rianne van Vredendaal, Léa Joffrin, Joachim Mariën.

**Writing – original draft:** Rianne van Vredendaal, Léa Joffrin, Antea Paviotti, Joachim Mariën.

**Writing – review & editing:** Rianne van Vredendaal, Léa Joffrin, Antea Paviotti, Claude Mande, Solange Milolo, Nicolas Laurent, Léa Fourchault, Pascal Baelo, Steve Ngoy, Martine Peeters, Ahidjo Ayouba, Jana Těšíková, Sophie Gryseels, Maha Salloum, Séverine Thys, Ynke Larivière, Gwen Lemey, Pierre Van Damme, Jean-Pierre Van Geertruyden, Hypolite Muhindo-Mavoko, Patrick Mitashi, Herwig Leirs, Erik Verheyen, Joachim Mariën.

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
