## [Decision Letter · Decision Letter 0]

11 May 2025

Assessing Ebola virus circulation in the Tshuapa province (Democratic Republic of the Congo): A One Health investigation of wildlife and human Interactions

PLOS Pathogens

Dear Dr. Vredendaal,

Thank you for submitting your manuscript to PLOS Pathogens. After careful consideration, we feel that it has merit but does not fully meet PLOS Pathogens's publication criteria as it currently stands. Therefore, we invite you to submit a revised version of the manuscript that addresses the points raised during the review process.

Please submit your revised manuscript within 60 days Jul 10 2025 11:59PM. If you will need more time than this to complete your revisions, please reply to this message or contact the journal office at plospathogens@plos.org. Please include the following items when submitting your revised manuscript:

We look forward to receiving your revised manuscript.

Kind regards,

Alexander Bukreyev, Ph.D.

Academic Editor

PLOS Pathogens

Kanta Subbarao

Section Editor

Editor-in-Chief

PLOS Pathogens

Editor-in-Chief

PLOS Pathogens

orcid.org/0000-0002-7699-2064

**Journal Requirements:**

At this stage, the following Authors/Authors require contributions: Rianne van Vredendaal, Léa Joffrin, Antea Paviotti, Claude Mande, Solange Milolo, Nicolas Laurent, Léa Fourchault, Douglas Adroabadrio, Pascal Baelo, Steve Ngoy, Papy Ansobi, Casimir Nebesse, Martine Peeters, Ahidjo Ayouba, Maeliss Champagne, Julie Bouillin, Jana Tĕšíková, Natalie Van Houtte, Sophie Gryseels, Maha Salloum, Freddy Bikioli, Séverine Thys, Jimmy Mpato, Ruben Ilonga, Henri Kimina, Ynke Larivière, Gwen Lemey, Pierre Van Damme, Jean-Pierre Van Geertruyden, Hypolite Muhindo-Mavoko, Patrick Mitashi, Herwig Leirs, Erik Verheyen, Guy-Crispin Gembu, and Joachim Mariën. Please ensure that the full contributions of each author are acknowledged in the "Add/Edit/Remove Authors" section of our submission form.

https://journals.plos.org/plospathogens/s/submission-guidelines#loc-parts-of-a-submission

5) We notice that your supplementary figure is uploaded with the file type 'Figure'. Please amend the file type to 'Supporting Information'. Please ensure that each Supporting Information file has a legend listed in the manuscript after the references list.

Potential Copyright Issues:

i) Figure 3. Please confirm whether you drew the images / clip-art within the figure panels by hand. If you did not draw the images, please provide (a) a link to the source of the images or icons and their license / terms of use; or (b) written permission from the copyright holder to publish the images or icons under our CC BY 4.0 license. Alternatively, you may replace the images with open source alternatives. See these open source resources you may use to replace images / clip-art:

ii) Figure 1. Please include in the figure legend (a) a direct link to the base layer of the map (i.e., the country or region border shape) ; and (b) a link to the terms of use / license information for the base layer image or shapefile. We cannot publish proprietary or copyrighted maps (e.g. Google Maps, Mapquest) and the terms of use for your map base layer must be compatible with our CC BY 4.0 license.

7) Thank you for stating "The cytochrome b sequences generated for host identification will be deposited in GenBank." Please note that, though access restrictions are acceptable now, your entire data will need to be made freely accessible if your manuscript is accepted for publication. This policy applies to all data except where public deposition would breach compliance with the protocol approved by your research ethics board. If you are unable to adhere to our open data policy, please kindly revise your statement to explain your reasoning and we will seek the editor's input on an exemption. Please be assured that, once you have provided your new statement, the assessment of your exemption will not hold up the peer review process.

8) Please ensure that the funders and grant numbers match between the Financial Disclosure field and the Funding Information tab in your submission form. Note that the funders must be provided in the same order in both places as well. "The European Federation of Pharmaceutical Industries and Associations, and the Coalition for Epidemic Preparedness Innovations" are missing from the Funding Information tab.

**Reviewers' Comments:**

Reviewer's Responses to Questions

**Part I - Summary**

Reviewer #1: van Vredendaal et al., Assessing Ebola virus circulation in the Tshuapa province (Democratic Republic of the Congo): A One Health investigation of wildlife and human interactions.

These data reported show the collection of specimens from small mammals and inhabitants of a village at the site of an Ebola virus (EBOV) disease outbreak in 2014. EBOV viral RNA was not detected in any of the samples tested. In contrast, ortoebolavirus IgG antibodies were detected in bats, rodents and individual villagers. The manuscript is informative and reports original data.

Reviewer #2: The manuscript PPATHOGENS-D-25-00366 from van Vredendaal et al. is exploring the role of small mammals in EBOV ecology and especially the trace of possible transmission from potential wildlife (bats) or peridomestic (rodents) reservoirs to the human population in and around the Inkanamongo village (DRC) where an Ebola outbreak occurred in 2014. The approach is claiming to respect the « One Health » concept by combining molecular and serological studies in human and animals and by screening sociological and environmental factors via questionnaires. While no trace of EBOV RNA (orthoebolavirus) was detected in any samples, animal and human populations were seroreactive to orthoebolavirus suggesting a former (or current) circulation of “EBOV-like” between them. Anti-orthoebolavirus IgG antibodies were present in wildlife blood samples (4% bats and 14,2% rodents). Among the human participants, 120/380 individuals (31,5%) had IgG antibodies against at least 1 orthoebolavirus antigen and 12 (3%) against 2 antigens of the same orthoebolavirus.

Although the molecular and serological analysis are seriously performed, trying to manage the inherent difficulties of the approach (absence of pos/neg controls...), the main weakness of the work consists in building a “One Health” chain of EBOV transmission between bats/rodents/human not sampled in the same year, nor in the same period of the year, sometimes in different locations

**Part II – Major Issues: Key Experiments Required for Acceptance**

Reviewer #1: Ln 199. Has the multiplex serological Luminex-based serological assay been fully validated in line with international standards? Would you include the Specificity / Sensitivity values and reproducibility index in testing bat sera.

Ln 285. Were any of the samples that tested positive confirmed by immunoblotting i.e. western blotting. If not, this should be undertaken.

Reviewer #2: As mentioned above, the main weakness of the work consists in building a “One Health” chain of EBOV transmission between bats/rodents/human not sampled in the same year, nor in the same period of the year, sometimes in different locations: indeed the animal samples were done in Inkanamongo village in May-June 2021 while the human samples were taken in December 2023. Concerning social interviews, a small part was done during the animal sampling in Inkanamongo village (May-June 2021) while the majority was performed in June 2022 in a larger Djerra sector (6 villages). Assuming the wildlife species seasonality, the current climate changes and possible landscape changes, as well as the probable versatility in human communities and occupation/behavior along the year, it is tedious to conclude holistically by comparing these different populations sampled/questioned at unrelated dates. Authors recognize themselves in lines 413-4 : « these discrepancies highlight the challenges of comparing seroprevalence data due to varying ecological settings, sampling dates, and seasons, all of which can influence the observed results ». Therefore :

• It would be interesting to have more information about the bat seasonality (migration?) in this region as well as the climate characteristics and potential landscape modification from 2021 to 2023 seasons. Authors mention in lines 428-30 “For example, the seasonal arrival of E. helvum around Inkanamongo coincides with fruit ripening, increasing contact with humans through hunting... » but do not give precision about the corresponding months and how they are matching or not with the sampling.

• Also, a complete “One Health” scope could explore, between bats/rodents and human, the potential role of livestock and domestic animals, particularly pigs, as intermediate hosts for Orthoebolavirus transmission. This is known for Reston virus and evidence of “Orthoebolavirus” seroprevalence in pigs has been shown in Sierra Leone, Guinea, Uganda and Ghana (doi.org/10.1093/infdis/jiy330 ; doi: 10.1111/tbed.13391; doi.org/10.1111/tbed.13822; doi.org/10.1292/jvms.22-0186 ; doi: 10.3201/eid3004.231034).

The social part is another subject: in the “Results” it appears over-sized and -detailed compared to the molecular and serological parts that are more compacted; in the “Discussion” it resembles a review discussing evidence from previous literature rather positioning the data of the present article. This results in a manuscript looking an assemblage of two independent parts. Indeed :

• most of the social observations are more descriptive than analysed in the context of the laboratory results. They could be shortened by integration in a table with only few words in the main text to highlight the important points.

• It would have been interesting to try to make a link between tested and interviewed individuals to see if their behaviour/occupation could be linked to sero-positivity. Unfortunately, testing (December 2023) was disconnected from the 40 interviews (May 2021). Even if some links could be recreated during testing, 2 years later, this would be only for a limited proportion of the 40 individuals, possibly too limited for statistics. For example is it possible to have more information about the activities, proximity to wildlife, of the 12/380 individuals found to be reactive against two antigens.

• Line 482-end: this is a long paragraph to explain the confusion about the origin and the index cas in the district. Authors mention themselves the poor reliability of several answers from the population for different reason. How are they managing this potential bias?

**Part III – Minor Issues: Editorial and Data Presentation Modifications**

Reviewer #1: Ln 85. I disagree with the statement that “detection of EBOV RNA proves infection at the time of sampling”. The presence of mRNA suggests an active infection. Viral RNA correlate with past infection. The text should be amended accordingly.

Ln 89. A citation should be included after the sentence ‘Substantial evidence……’ One suggestion is Luis et al. A comparison of bats and rodents as reservoirs of zoonotic viruses: are bats special? Proc Biol Sci. 2013 Feb 1;280(1756):20122753. doi: 10.1098/rspb.2012.2753. PMID: 23378666; PMCID: PMC3574368.

Ln 105: An additional reason is that many studies focus on catching and sampling healthy bats. This in proven to be unsatisfactory. The text should be amended accordingly.

Ln 116. Additional topics also include collecting bat guano for use as fertilizer, religious practices (preying) in caves habituated by bats and hunting bushmeat.

Ln 244. Did the social science study include the following: medical and travel history.

Ln 265. A structured questionnaire should be included in a supplementary file.

Ln 353. Does the consumption of wild meat by villagers include the consumption of bats? If so, the text should be amended accordingly.

Table 1 and Fig.1. provide supporting information and should be considered as Supplementary Files.

Reviewer #2: • Is pooling samples susceptible to render the test less sensitive (false negative)? Alternatively, is the nature of sampling possibly involved? Non-invasive samples in bats, not suitable organs in rodents. Marburg virus was isolated in bat spleen. Could this be discussed?

• It will be important to precise in which recombinant system the proteins (GP, NP, VP40) have been produced since this may play a substantial role in the recognition by Abs (glycosylation, etc). For example, in lines 280-82, it is surprising to find bat cross-reacting with the GP or the VP40 proteins of different orthoebolaviruses and not with the NP which is globally less divergent between viruses. As well, lines 282-84 it is surprising that no bat recognise different antigens of the same orthoebolavirus. Would it be linked to the lower quantity/quality of one antigen? For example, the NP antigen seems slightly less reactive compared to VP40 and GP. Is it linked to its production or to its lower immunogenicity and more rapid decline of humoral immunity? Authors evoke this last controversial point in the literature in lines

460-3. The notion of the quality (production) of the antigen used in the test must be also discussed.

• For rodents, authors mentioned “trapping in the villages”. Is it possible to precise if the traps were peri-domestic or in houses? The more seropositive rodent is Lophomorus rita (15/38). Interesting to observe that some L. rita samples can recognize different proteins of the same orthoebolavirus, even if from different species (EBOV and BDBV). Could Lophomorus rita be a candidate as filovirus reservoir compared to Mus or Rattus sp. Is it found in or around houses ? What are its preference in term of food, roosts, behaviour. During the interviews, did population mention or not the consummation of wild rodents?

• Among IgG reaction for at least two Orthoebolavirus, it is interesting to note the coupled EBOV/BDBV since BDBV was responsible for an outbreak In DRC in 2012 with fatal cases.

Minor comment

• Is it possible to decrease the intensity of green in fig 1 (lower square down right) to clearly distinguish the points indicating capture sites.

PLOS authors have the option to publish the peer review history of their article (what does this mean? ). If published, this will include your full peer review and any attached files.

**Do you want your identity to be public for this peer review?** For information about this choice, including consent withdrawal, please see our Privacy Policy .

Reviewer #1: No

Reviewer #2: No

**Figure resubmission:**

**Reproducibility:**



---

## [Decision Letter · Decision Letter 1]

8 Sep 2025

PPATHOGENS-D-25-00366R1

Assessing Ebola virus circulation in the Tshuapa province (Democratic Republic of the Congo): A One Health investigation of wildlife and human interactions

PLOS Pathogens

Dear Dr. van Vredendaal,

Thank you for submitting your manuscript to PLOS Pathogens. After careful consideration, we feel that it has merit but does not fully meet PLOS Pathogens's publication criteria as it currently stands. Therefore, we invite you to submit a revised version of the manuscript that addresses the points raised during the review process.

Please submit your revised manuscript within 30 days Nov 07 2025 11:59PM. If you will need more time than this to complete your revisions, please reply to this message or contact the journal office at plospathogens@plos.org. Please include the following items when submitting your revised manuscript:

We look forward to receiving your revised manuscript.

Kind regards,

Kanta Subbarao

Section Editor

PLOS Pathogens

Kanta Subbarao

Section Editor

PLOS Pathogens

Sumita Bhaduri-McIntosh

Editor-in-Chief

PLOS Pathogens

orcid.org/0000-0003-2946-9497

Michael Malim

Editor-in-Chief

PLOS Pathogens

orcid.org/0000-0002-7699-2064

**Additional Editor Comments:**

Editor:

Please summarise your response to Reviewer 1's question regarding the multiplex serological Luminex-based serological assay in a few sentences and add them to the methods section.

**Reviewers' Comments:**

Reviewer's Responses to Questions

**Part I - Summary**

Reviewer #2: Authors have done a clear effort to precisely respond to the reviewer's comments. Most of the weaknesses outlined by the reviewers have been (1) frankly recognised when impossible to clear them or (2) responded and eventually amended in the manuscript wherever possible to do. In both cases, an extended discussion with relevant references shows that authors are expert in the topic.

The response concerning the patchworking of dates, seasons and places of sampling is based on the limitation of studies due to COVID and regional instability and the limited importance of matching human and animal sampling period for serology since antibodies persist in blood. The latter is possible on the one hand, but on the other hand the litterature is also scaterred with examples where the antibody level can stronlgy evolve in 2,5 years (May 2021, December 2023)...

A strong point of the work is the tentative to respect the One Health context by focusing a specific village (Inkanamongo) even if at different periods. Also, the text has benefited of a interesting and substancial addition the seasonal aspect with precise examples underlining a link between consumption of freshly roosting bats and infection.

**Part II – Major Issues: Key Experiments Required for Acceptance**

Reviewer #2: All questions in this part have been responded for the satisfaction of Reviewer n°2

**Part III – Minor Issues: Editorial and Data Presentation Modifications**

Reviewer #2: All questions in this part have been responded for the satisfaction of Reviewer n°2

PLOS authors have the option to publish the peer review history of their article (what does this mean? ). If published, this will include your full peer review and any attached files.

**Do you want your identity to be public for this peer review?** For information about this choice, including consent withdrawal, please see our Privacy Policy .

Reviewer #2: No

**Figure resubmission:**
---

## [Editor Report · Decision Letter 2]

14 Oct 2025

Dear Ms van Vredendaal,

We are pleased to inform you that your manuscript 'Assessing Ebola virus circulation in the Tshuapa province (Democratic Republic of the Congo): A One Health investigation of wildlife and human interactions' has been provisionally accepted for publication in PLOS Pathogens.

Best regards,

Kanta Subbarao

Section Editor

PLOS Pathogens

Kanta Subbarao

Section Editor

PLOS Pathogens

Sumita Bhaduri-McIntosh

Editor-in-Chief

PLOS Pathogens

orcid.org/0000-0003-2946-9497

Michael Malim

Editor-in-Chief

PLOS Pathogens

orcid.org/0000-0002-7699-2064
---

## [Editor Report · Acceptance letter]

21 Nov 2025

Dear Ms van Vredendaal,

We are delighted to inform you that your manuscript, " 

Assessing Ebola virus circulation in the Tshuapa province (Democratic Republic of the Congo): A One Health investigation of wildlife and human interactions," has been formally accepted for publication in PLOS Pathogens.

Best regards,

Sumita Bhaduri-McIntosh

Editor-in-Chief

PLOS Pathogens

orcid.org/0000-0003-2946-9497

Michael Malim

Editor-in-Chief

PLOS Pathogens

orcid.org/0000-0002-7699-2064